

# Algorithm for continual monitoring of fog life cycles based on geostationary satellite imagery as a basis for solar energy forecasting

Babak Jahani[1,2], Steffen Karalus[3], Julia Fuchs[1,2], Tobias Zech[3], Marina Zara[1,2], Jan Cermak[1,2]

[1]Institute of Meteorology and Climate Research, Karlsruhe Institute of Technology (KIT), Karlsruhe, Germany.
5   [2]Institute of Photogrammetry and Remote Sensing, Karlsruhe Institute of Technology (KIT), Karlsruhe, Germany.
[3]Fraunhofer Institute for Solar Energy Systems ISE, Freiburg, Germany.

*Correspondence to*: Babak Jahani (babak.jahani@kit.edu; b.jahani@sron.nl); Jan Cermak (jan.cermak@kit.edu)



**Abstract.** Detection and monitoring of fog and low stratus (FLS) is particularly important in the context of photovoltaic power production, as FLS, unlike moving clouds, can persist longer and impact larger regions simultaneously, making regional power grid balancing harder. Although photovoltaic power production is limited to daytime hours, its short-term forecasting (especially during the early hours of the day) in the context of high PV penetration systems grid operation, benefits from a complete knowledge of FLS life cycle. As this life cycle usually begins at night and ends during the day, a day-night consistency in the algorithms used for monitoring FLS is required. This study presents an algorithm for detection of FLS over Europe based on the infrared bands of the SEVIRI (Spinning Enhanced Visible and InfraRed Imager) instrument onboard the Meteosat second generation geostationary satellites. As the method operates based on the SEVIRI infrared observations only, it is expected to be stationary in time and thus can provide a coherent and detailed view of FLS development over large areas over the 24H day cycle. The algorithm is based on a gradient boosted trees machine learning model that is trained with ground truth observations from Meteorological Aviation Routine Weather Reports (METAR) stations and the SEVIRI observations at bands cantered at 8.7, 10.8, 12.0 and 13.4 μm wavelengths. The METAR data used here comprises a total number of 2,544,400 datapoints spread over the winters (i.e., 1st of September to 31st of May) of the years 2016-2022 and 356 locations across Europe. Among them, the datapoints corresponding to 276 stations and the winters of 2016-018 and 2019-2021 (~45% of all datapoints) were used to train the algorithm. The remaining datapoints comprise four independent datasets which were used to validate the algorithm's performance and applicability to the time spans and locations in the study area (i.e., Europe) that extend beyond particular locations and time spans covered by the datapoints used for training the algorithm. Additionally, the algorithm's accuracy at the locations of METAR stations with that of the stablished state-of-the-art daytime FLS detection algorithm Satellite-based Operational Fog Observation Scheme (SOFOS). Validation of the algorithm against the METAR data, showed that the algorithm is well suited for detection of FLS. Specifically, the algorithm is found to detect FLS with probabilities of detection (POD) ranging from 0.70 to 0.82 (for different inter-comparison approaches), and false alarm ratios (FAR) between 0.21 and 0.31. These numbers are very close to those achieved by SOFOS for discriminating the FLS from other sky conditions at the tested locations and time spans. These results also showed that the technique's applicability in the study region extends beyond the particular locations and time spans covered by the datapoints used for training the algorithm.

# 1 Introduction

Fog and low stratus (FLS) are both persistent aggregations of water particles in liquid and/or solid phases, i.e., clouds close to the Earth's surface. As the cloud-base-height is the only real difference between the two (fog: touching the ground; low stratus: above ground), they are frequently treated together as a single category from the satellite perspective (i.e., FLS). FLS influences various aspects of life on the Earth: on the one hand it may act as source of water supply (e.g. Lehnert et al., 2018; Shanyengana et al., 2002), as well as a modifier in the global climate system (Vautard et al., 2009). On the other hand, it causes large economic costs in transport, health and energy sectors (Köhler et al., 2017; Pérez-Díaz et al., 2017). For these



reasons, continuous (and ideally near real-time) detection of FLS and monitoring of its spatiotemporal patterns and developments is of essential importance. One important application lies in the usage solar energy. With the rising share of electricity generated from photovoltaic (PV) systems, the prediction of current and future solar irradiance and, hence, PV power output becomes crucial for the operation of modern power grids. In the context of PV power production, detection and monitoring of FLS is particularly important as FLS, unlike moving clouds, can persist longer and impact larger regions simultaneously, making regional power grid balancing a challenging task.

Geostationary satellites have an outstanding potential for detection and monitoring of FLS. That is because they continuously scan the Earth from the same angle over the 24H diurnal cycle, thus providing spatiotemporally coherent spectral images of the Earth. To exploit this potential, to this day several FLS detection methods have been developed and applied (e.g., Cermak & Bendix, 2007, 2008; Egli et al., 2018; Ellrod, 1995; Nilo et al., 2018; Underwood et al., 2004), which detect FLS by performing a sequence of spectral and/or spatial tests on the satellite measurements. Although these methods have proven to be useful exploring the FLS characteristics (e.g., Cermak et al., 2009; Egli et al., 2017; Pauli, Cermak, & Andersen, 2022; Pauli, Cermak, & Teuling, 2022), they usually consist of separate daytime and nighttime schemes, making use of spectral channel characteristics particular for either night or day. This, however, makes the continuous monitoring and forecasting of FLS life cycle and occurrence at the critical of the day (i.e., sunrise) impossible. For example, a FLS detection method may operate based on the shortwave reflectances, as FLS covered regions typically have high reflectivity and smooth texture in the visible spectral region (Lee et al., 2011). But as no reflectance data is available during nighttime a whole other algorithm will be required to be applied over nighttime. FLS can also be detected by testing the difference between the brightness temperatures in Medium (MIR) and Long (LIR) wavelength Infrared bands (typically referred to radiation at wavelengths between 3-8 and 8-15 μm, respectively) against threshold values. That is because as the FLS droplets are typically smaller than those of other cloud types. As a result, for them, the contrast between the emissivities at MIR and LIR bands is greater compared to other clouds (Hunt, 1973). However, as MIR has a solar component during daytime, application of this technique requires daytime and nighttime separation. A stable FLS detection approach applicable during day and night would have to solely depend on observations at the LIR bands. As a complete knowledge of FLS event development in space and time is needed as a basis for short-term forecasting, e.g. in the context of grid operation (in case of high PV penetration systems), the main objective of the present study is to develop and validate a single fully diurnal FLS detection algorithm for Europe based on geostationary satellite observations. The guiding hypothesis of the present study is that FLS can be discriminated from cloud-free regions and non-FLS clouds based on spectral LIR data with an accuracy comparable with the existing state-of-the-art daytime FLS detection algorithms.

Satellite detection of FLS using LIR bands only is a quite challenging task and historically believed to be a rather impossible one (Güls & Bendix, 1996). That is mainly because as FLS emit at temperatures very close to that of the Earth's surface. Nonetheless, Andersen & Cermak (2018) succeeded developing a method for FLS detection over the Namib desert (located in Southern Africa) which operates solely based on the brightness temperatures measured at the LIR bands of the Spinning Enhanced Visible and InfraRed Imager (SEVIRI) sensor on board the Meteosat Second Generation (MSG) spacecraft.



Specifically, they discriminated FLS from cloud-free land by analysing the spatial structure (texture) of images constructed from the difference between the brightness temperatures measured at 12.0 and 8.7 μm SEVIRI channels. The main underlying assumptions associated with this method are i) the 12.0-8.7 μm values are typically greater for clouds than for land surface, and ii) the 12.0-8.7 μm calculated for bare soil shows a higher level of spatial heterogeneity compared to that of the smooth top surfaces of FLS. Their success in developing their algorithm shows that although small, there are distinct

differences between the spectral signatures of a clear-land and FLS in the LIR region, which makes it possible to distinguish them from each other. Nevertheless, this method is applicable to regions where the mentioned assumptions are met (i.e., deserts and possibly arid regions) and cannot be easily ported to the much more complex conditions of Europe. Therefore, to this date, no approach for the continual, 24H detection of FLS over Europe exists.

    This study presents and validates a single machine-learning (ML) algorithm for detection of FLS over land across Europe,

based on Meteosat-SEVIRI bands cantered at 8.7, 10.8, 12.0 and 13.4 μm wavelengths. Furthermore, it provides an intercomparison between the accuracy of this method and that of the established state-of-the-art daytime FLS detection algorithm Satellite-based Operational Fog Observation Scheme (SOFOS; Cermak, 2006; Cermak & Bendix, 2008). The method presented here is based on a gradient boosted trees (XGBoost) ML model that is trained with observations from METeorological Aviation routine weather Reports (METAR) stations and the LIR observations of SEVIRI on board

Meteosat-10 and Meteosat-11 platforms. As the method presented here operates solely based on the spectral data in the LIR region, it is applicable during the 24H diurnal cycle over European lands. Thus, it is suitable for continuous monitoring of FLS over Europe and can reveal a detailed view of the diurnal and spatial patterns of FLS over Europe. This can be of essential value for analysis of FLS occurrence and related processes as well as improvement of methods for PV power forecasting. It can also serve as a precondition for statistics-based nowcasting.

**2 Data and Methods**

**2.1 Satellite observations**

    SEVIRI is a multi-purpose scanning (passive) radiometer onboard MSG geostationary satellites operated by the European Organization for Exploitation of Meteorological Satellites (EUMETSAT; Aminou, 2002). This instrument has been in operation by EUMETSAT at 0° latitude and longitude at the altitude of ~35800 km above mean sea level since early 2004

and is intended to remain in service at 0° until 2033 (expected lifetime of the last MSG mission, i.e., Meteosat-10). SEVIRI measures upwelling radiance at the top of the atmosphere (TOA) for the Earth's whole disk covering Europe, the North Atlantic, and Africa in 12 spectral channels in 15-minute intervals (12 minutes scanning time followed by 3 minutes of processing time). Eleven of the SEVIRI channels are narrow-band channels which are spread over the spectral region between 0.56 (visible) to 14.4 (LIR) μm and have a sampling spatial resolution of 3x3 km at nadir (they are referred to as

low-resolution channels). The remaining one is a broadband High Resolution Visible channel (0.6 - 0.9 μm; referred to as HRV) with a sampling resolution of 1 km at nadir. The level-1.5 product of SEVIRI (Hanson & Mueller, 2004; Schmetz et



al., 2002; Tranquilli et al., 2016) consists of geolocated, radiometrically pre-processed TOA upwelling radiances observed by SEVIRI (spacecraft specific effects have been removed) at its 12 channels. For the present study, the level 1.5 radiances measured by SEVIRI instruments at 0 degree at channels cantered at 0.6, 0.8, 1.6, 3.9, 8.7, 10.8, 12.0 μm wavelengths were
utilized. This data was acquired from EUMETSAT for region covering Europe and north Africa and includes the winter periods (i.e., spans 1st of September to 31st of May) between the years 2016 and 2022. This data was measured by the SEVIRI instruments onboard MSG-3 and MSG-4 platforms. The acquired data was then calibrated and converted to reflectance ($\rho$; for 0.6, 0.8, 1.6 μm channels) and brightness temperature ($BT$; for 3.9, 8.7, 10.8, 12.0 and 13.4 μm channels) following the procedure explained in (EUMETSAT, 2017) using the satpy python package (Martin et al., 2021).

**2.2 Ground-based observations**

The METAR (METeorological Aviation Routine Weather Reports) comprise the reports of weather measurements performed at a network of meteorological stations located at the airports across the world. The primary objective of these measurements is for aviation safety, but they also have an application in meteorology. The METAR parameters cloud base height (CBH; m), sky cloud cover (CC; okta) and horizontal visibility (HV; km) for the period 1st of September 2016 to 31st
of May 2022 were acquired and used in the present study to identify the FLS conditions based on ground-based observations. These parameters are retrieved either automatically (instrumentation is not standardized) or estimated by human observers at each station at temporal frequencies of 1 hour or less. The temporal frequency of the observations performed at these stations is related to the weather conditions, airport size and traffic. In particular, stations located at large, and busy airports are equipped with automatic devices and tend to have a higher frequency of observations compared to small airports. Also,
under weather conditions violating the flight safety such as fog occurrence or extreme weather conditions, the frequency of the observations at these stations is increased to ensure the flight safety.

The acquired data was first subjected to a quality control procedure to filter out the stations with unreliable observations. Specifically, the stations not meeting one of the following conditions were considered as unreliable and were discarded from the analysis: 1) average number of observations per hour greater than 1.5 for the data period, 2) average number of empty
observations less than 0.05 during the data period, or 3) continues data gaps less than one month. The underlying assumption for defining these tests is that the data from the stations with rather homogeneous frequency of observations is more likely to have a reliable accuracy and consistency. The quality control thresholds applied here have been defined empirically as such they are restrictive enough to filter out the stations with unreliable observations, and yet relaxed enough to get a reasonable number of temporally homogeneous observations for further analysis. By applying the abovementioned filters, 591 out of
947 stations were discarded. The locations of the remaining 356 stations were spatially matched with the SEVIRI grid using the smallest distance between the centre of each SEVIRI pixel and the station location. For the temporal matching, the METAR times of observation were first rounded up to the closest 15-minute intervals ($t_M$) of SEVIRI. Then, the SEVIRI images with the timestamps $t_M$+15minute were considered as the temporally matched images, as the actual SEVIRI scanning time for the study area is much closer to the end of each 15-minute interval. The location of these stations is illustrated in



Fig. 1. As can be inferred from this figure, the stations selected are spread across Europe (plus four in north Africa), covering a wide range of regions with various surface types, altitudes, meteorological conditions.

**Figure 1. Geographical location of the selected ground observation (METAR) sites. The horizontal color bar indicates the elevation above mean sea level for the study area (m; obtained from EUMETSAT: last access: 16.01.2023 - 13:36 GMT+1). The filled circles show the geographical locations of the stations included in the "train" and "test₁" datasets. The triangles illustrate the geographical locations of the stations included in the datasets "test₂" and "test₃", (see Sect. 2.4 for more information). The locations of the stations included in test4 dataset are shown with plus ("+") signs. The vertical color bar indicates the mean annual**



FLS occurrence frequency ($\bar{f}$) at the location of the station as indicated according to the procedure explained in Sect. 2.4 of the
manuscript based on the modified ground-based FLS labels (before selecting the confident labels).

## 2.3 SOFOS

As a reference for the newly developed algorithm, the well-established and well-validated satellite-based the Operational

Fog Observation Scheme (SOFOS, Cermak 2006, Cermak & Bendix 2008) was used. A daytime-only technique, this

approach was developed specifically for MSG SEVIRI, and makes use of the visible and mid-infrared and thermal infrared

channels in a series of threshold tests. These include per-pixel spectral tests as well as tests applied on spatially coherent and

distinct areas of pixels.

## 2.4 FLS labels

The CBH, CC and HV measurements performed at the locations of the selected stations were used for the ground-based

identification of FLS occurrences. To this aim, the case of fog and homogeneous low stratus conditions were first determined

separately and then the two were combined to produce the ground-based FLS labels. Specifically, the data points with HV

less than 1 km were labelled as fog and those with CBH of the lowest cloud layer less than 1 km and CC greater than 5 octas

were labelled as homogeneous low stratus conditions (Egli et al., 2017). The data points for which either fog or low stratus

was observed were labelled as FLS positive. Data pints with no reported values for at least one of the parameters HV, CBH

or CC were labelled as undefined. Data points for which neither fog nor low status was observed were labelled as FLS

negative. The FLS labels produced this way indicate whether FLS is observed from the perspective of ground-based

observer/instrument. However, these labels may not be consistent with what is seen from the space, as the satellite's field of

view (FOV) may be blocked by layers of non-FLS clouds passaging through its line of sight (LOS). Another issue is that the

FLS negative labels indicated as above do not specify the sky condition (cloudy or cloud-free-land) in the satellite's FOV.

To cope with these two issues, the non-FLS clouds blocking the satellite's LOS were identified using the SEVIRI spectral

measurements. Particularly, a series of spectral tests were performed using the collocated SEVIRI $\rho_{0.6}$, $\rho_{1.6}$, $BT_{3.9}$, $BT_{8.7}$,

$BT_{10.8}$, $BT_{12.0}$ and $BT_{13.4}$ data corresponding to each data point. These tests are the same those used in the SOFOS algorithm

for classification of the SOFOS classes "water", "Cirrus" and "ice" clouds (Cermak & Bendix, 2008) and can rather

confidently discriminate non-FLS clouds from FLS and cloud-free land. Afterwards, the abovementioned FLS labels were

modified according to the non-FLS cloud observations as follows: among the data points labelled as FLS negative based on

the ground data, those which were non-FLS-cloud-free and non-FLS-cloud-covered based on the SEVIRI data were

classified as "clear-sky" and "non-FLS-cloud" conditions, respectively. As for the data points which were labelled as FLS

positive, those identified as non-FLS-cloud-free and non-FLS-cloud-covered were classified as "FLS" and "multi-layer"

(FLS under a non-FLS cloud) conditions, respectively. Worth mentioning that as the non-FLS cloud screening procedure

explained above uses the SEVIRI $\rho_{0.6}$, $\rho_{1.6}$ data, its application is limited to daytime (i.e., solar zenith angles $\leq 80°$). For this

reason, despite the availability of both the SEVIRI and METAR over day and night, the FLS labels derived here are limited



to the daytime only. It should also be noted that, although the procedure applied here for detection of "non-FLS-cloud" contaminated cases is rather legit, there times that it fails to detect them. As a result of this failure, some of the labels may have been mistakenly modified. Specifically, some multi-layer and non-FLS-cloud conditions may have been mistakenly labelled as FLS and clear-sky (and vice versa), respectively. Nevertheless, initial tests for fitting a ML model over the data

showed that application of the abovementioned procedure for detection of non-FLS-clouds is very essential for optimizing the model's performance in identification of the FLS conditions.

In the next step, the entire data was split into five datasets, namely, "train", "test₁", "test₂", "test₃" and "test₄". Table 1 provides a summary about the characteristics of these datasets and Fig. 1 shows the location of the stations included in each of them. The train dataset was used for training the ML algorithm. test₁ dataset was intended to evaluate the performance of

the trained model over locations seen by the model during training, but at time spans beyond that of the data seen by the model during training. The datasets test₂, test₃ and test₄ were intended for quantifying the accuracy of the trained model over locations which were never seen by the model during the whole study period. To split the data to the five datasets, the annual frequency of FLS occurrence at each station was first determined by calculating the ratio of the daytime FLS positive labels to all valid labels in METAR per winter year (i.e., 2016-2017, 2017-2018, …). In the second step, the annual ratios were

averaged over the study period to obtain the mean annual frequency of FLS occurrence at each station ($\bar{f}$, range of the values obtained: 0.00-0.12). In the third step, the stations were grouped into 23 bins based on their $\bar{f}$ values (bin widths: 0.005). The entire data corresponding to the stations falling in the first two $\bar{f}$ bins (i.e., $\bar{f}$<0.01) were considered as the test₄ dataset (59 stations). This dataset was intended to evaluate the algorithms' performance at location where FLS rarely occurs. In the fourth step, from each of the 21 remaining $\bar{f}$ bins, one random station per bin was selected to construct the test₂ and test₃

datasets. Among all the datapoints corresponding to these 21 stations, those covering the winters (September to May) of 2018-2019 and 2021-2022 comprise test₃ dataset and the remaining ones comprise test₂ dataset. These two datasets are considered for validating the algorithms at locations other than those used for training the ML model. As can be understood, the difference between the two is time span covered by them. In particular, the time spans covered by the dataset test₂ are the same as those of the dataset used for training, and test₃ dataset covers time spans beyond those covered by the train dataset.

Lastly, the test₁ and train datasets were then constructed by splitting the data corresponding to the 276 stations which were not allocated to test₂, test₃ or test₄. Specifically, the data points corresponding to the winters (September to May) of 2018-2019 and 2021-2022 comprised test₁ dataset, and those corresponding to the winters of 2016-2017, 2017-2018, 2019-2020 and 2020-2021 comprise the train dataset. As can be inferred, the data points corresponding to the stations with $\bar{f}$ less than 0.01 were not included in the train dataset. This data was excluded from the train dataset to decrease the imbalance between the FLS positive and negative cases. That is because the initial tests of the algorithm indicated that the data corresponding to

the stations with $\bar{f}$ less than 0.01 in the train dataset vastly increases the imbalance between FLS positive and negative cases, which lead to a decrease in the algorithm's skill for detecting the FLS cases. This data was, however, included in the analysis as test₃ dataset to evaluate the algorithm's performance at places that FLS rarely occurs.



Afterwards, the quality of the modified FLS labels was assessed and the case of "no confidence" and "semi-confident" FLS were removed from the four datasets. This was done by comparing the labels corresponding to each timestamp at each station with that of its previous and next timestamps at the same station. Particularly, three consecutive data points at the same location were chosen each time. If the time difference between the first and third data points was greater than 1hr, or the FLS label for neither first nor third data point were the same the middle one, the quality flag for the middle data point was set to "no confidence". If the FLS label for one of the first or third data points was the same as that of the middle one, then the quality flag for the middle data point was set to "semi-confident". Lastly, if the FLS label for both the first and third data points was the same as that of the middle one, then the quality flag for the middle data point was set to "confident". This quality control procedure was considered to select the data-points with homogeneous sky condition and to cope with the uncertainties associated with spatio-temporal matching of the SEVIRI data with the ground station data, the coordinate system used for geolocation of SEVIRI pixels, and coordinates of the ground stations. The SEVIRI pixels have a spatial resolution of about 3km at nadir which results in a resolution between 4 to 8km over Europe. Plus, there is up to 0.3km of uncertainty associated with the procedure applied by EUMETSAT for georeferencing SEVIRI images. Additionally, the satellite is not exactly stationary on the geostationary ring and its true position varies slightly with time because all the satellites are drifting from their nominal position. On the other hand, the ground stations are not exactly located at the centre of the SEVIRI pixels, and some are indeed on the borders of the pixels. Also, there may exist occasions where the METAR location within the pixel has a different sky cover compared to the majority of the pixel area. This can be the case with pixels contaminated by the edges of non-FLS clouds or cloud layers which are forming, but are not yet fully developed (cf. Jahani et al., 2020, 2022; Koren et al., 2007). The sky conditions for the data points surviving this quality control procedure are assumed to be homogeneous at the subpixel level.

Next, the case of "multi-layer" conditions were relabelled as "non-FLS-cloud" for the $test_1$, $test_2$, $test_3$, and $test_4$ datasets but were removed from the train dataset. They were removed from the train dataset to have excluded the cases that can potentially be ambiguous and thus have the ML model trained on a dataset which contains a clear discrimination between "non-FLS-cloud" and the two other classes (i.e., "clear-sky" and "FLS"). The METAR FLS labels generated in this way (hereafter referred to as MFL) serve as the ground truth and will be used for the model development and validation purposes. A summary of the data included in the final four datasets is given in Table 1. A schematic description of the process followed in this section for creation of the MFL included all the five datasets is provided in Fig. A1. As this table shows, the datasets train, $test_1$, $test_2$, and $test_3$ contained an overall number of 1168921, 644804, 142641, and 588034 data points, respectively. This table also shows the number of non-FLS cases (i.e., clear-sky and non-FLS-cloud classed combined) is substantially higher than that of the FLS cases. Nevertheless, the faction of FLS cases is in the same range for the datasets train (3.8 %), $test_1$ (3.3 %), $test_2$ (4.7 %), and $test_3$ (4.4 %) whereas that of $test_3$ (0.2 %) is considerably lower than that of the other three datasets.






**Table 1. Summary of the data contained in the "train", "test₁", "test₂", and "test₃" datasets. The values given in this table are based on the final datasets (i.e., they are derived based confident FLS labels).**

| Dataset | Time span | Stations | Total data points | Label frequency in the entire dataset (%) | | |
|---------|-----------|----------|-------------------|-------------|-----|----------------|
| | | | | clear-sky | FLS | non-FLS-cloud |
| **train** | 2016-2017 | 276 | 1168921 | 35.6 | 3.8 | 60.6 |
| | 2017-2018 | | | | | |
| | 2019-2020 | | | | | |
| | 2020-2021 | | | | | |
| **test₁** | 2018-2019 | 276 | 644804 | 33.4 | 3.3 | 63.3 |
| | 2021-2022 | | | | | |
| **test₂** | 2016-2017 | 21 | 95818 | 32.3 | 4.7 | 63.0 |
| | 2017-2018 | | | | | |
| | 2019-2020 | | | | | |
| | 2020-2021 | | | | | |
| **test₃** | 2018-2019 | 21 | 46823 | 33.9 | 4.4 | 61.7 |
| | 2021-2022 | | | | | |
| **test₄** | 2016-2022 | 59 | 588034 | 39.1 | 0.2 | 60.7 |

## 2.5 Machine learning algorithm


In the present study, a XGBoost (gradient boosted trees) model was developed for FLS detection over Europe based on SEVIRI observations in the LIR (Long InfraRed) spectral region. XGBoost is a well-tested supervised ML (machine learning) technique which has been successfully applied to many regression and classification problems. It extracts the



nonlinear relationships between sets of input variables/features and a target output variable through constructing an ensemble
of sequentially built regression trees (also referred to as weak learners) to the data. The regression trees are added one at a
time to the ensemble to correct for the deficiencies in the previous model and minimizing a specified loss function. All these
trees together construct a powerful statistical model referred here to as XGBoost. For this model, the prediction is performed
by summing over all the regression trees. Once the model is trained, the role of each input feature in constructing the boosted
decision trees within the model is indicated based on a metric referred to as feature importance. The more an input feature is
used to make key decisions with decision trees, the higher its importance. For more information about XGBoost please refer
to (Friedman, 2001; Mitchell & Frank, 2017; Natekin & Knoll, 2013). The open source XGBoost python implementation
(https://xgboost.readthedocs.io/en/stable/index.html, last access: 28.03.2023) was used in the present study.

The XGBoost model was trained to predict the labels "FLS", "non-FLS-cloud", and "cloud-free" based on a set of input
variables generated from the channel and channel combinations $BT_{12.0}$, $BT_{8.7}$ - $BT_{12.0}$, $BT_{10.8}$ - $BT_{12.0}$, and $BT_{12.0}$ - $BT_{13.4}$ plus
the standard deviation of each of these variables in a spatial window sized 3x3 pixels with the central pixel being the target
pixel. The pixel values of the mentioned variables contain information about the cloud presence, cloud top height, phase,
particle radius for the area falling within the limits of the target pixel and their standard deviations summarize the
heterogeneity of this information over the 3x3 pixel area around the target pixel. The standard deviations were considered
because the different land and cloud types tend to show different degrees of spectral and spatial heterogeneity.

The set of input variables mentioned above were selected by analysing the spectral SEVIRI LIR data and the corresponding
the MFLs included in the "train" dataset with the objective of creating the simplest model possible that can capture general
differences between FLS and the two other classes (i.e., non-FLS-cloud and cloud-free). Here, simplest model is defined as
an accurate model (according to Eqs. (1-7)) capable of detecting FLS based on minimal spectral and spatial input data. To
select the most relevant input features for discriminating FLS from the two other classes (i.e., those that reduce the error the
most), the data corresponding to the LIR channels $BT_{8.7}$, $BT_{9.7}$, $BT_{10.8}$, $BT_{12.0}$ and $BT_{13.4}$ along with all the 10 possible
combinations which can be derived from subtracting them from one another were tested. This was done by applying the
feature selection technique referred to as "backwards elimination". To this aim, a XGBoost model was initially trained to
predict FLS labels using all the input features. The features were then iteratively removed (one feature was dropped per
iteration) based on their low gains in fitting (feature importance), leading to a series of models, totalling 15. The accuracy of
these 15 models in detection of the FLS was evaluated based on Eqs.(1-7) given in Sect. 2.6. The simplest combination of
input features was then selected by comparing the accuracy of the models 2 to 15 with that of the first model (the most
complex one was considered as the reference). This revealed accuracy drops gradually as features decrease, with a
significant drop when a key feature is removed. The model just before this drop signifies the minimal essential feature set. In
this way, the variables $BT_{12.0}$, $BT_{8.7}$ - $BT_{12.0}$, $BT_{10.8}$ - $BT_{12.0}$, and $BT_{12.0}$ - $BT_{13.4}$ were chosen as the most essential inputs for
the XGBoost model. Additional tests also indicated that including the standard deviations of these variables in a 3x3 pixel



area around the target pixel helps increasing the accuracy of the model and for this reason, they were considered as input features to the model.

The hyperparameters of the model were tuned to optimize the model's performance and avoid overfitting. This was done on the basis of the statistical indicators explained in Sect. 2.6. To this aim, a grid-search was performed to find the optimum

values of the XGBoost hyper parameters learning rate (0.3), maximum depth (5), minimum child weight (1), number of regression trees (100), alpha (0), and lambda (1). The XGBoost model trained this way was then applied to the entire SEVIRI scenes acquired in the present study (see Sect. 2.1), including the SEVIRI pixels over water. The metrics used for selection of the hyperparameters are presented in Sect. 2.6.

## 2.6 Validation

The skills of the newly proposed ML and the established SOFOS FLS detection algorithms were quantified and compared by validating them versus the MFLs corresponding to the "train", "test$_1$" and "test$_2$" datasets. The validation for each dataset was performed by calculating the statistical indicators *Probability of Detection* (*POD*), *False Alarm Ratio* (*FAR*), *Probability of False Detection (PFD)*, *Critical Success Index* (*CSI*), *Accuracy (ACC), Bias Score* (*BS*), and *distance from optimal point (d)* as follows:

$POD = N_H / (N_H + N_M)$                                                  (1)

$FAR = N_F / (N_H + N_F)$                                                   (2)

$PFD = N_N / (N_F + N_N)$                                                  (3)

$CSI = N_H / (N_H + N_M + N_F)$                                            (4)

$ACC = (N_H + N_N) / (N_H + N_N + N_F + N_M)$                           (5)

$BS = (N_H + N_F) / (N_H + N_M)$                                         (6)

$d \ = ((POD\text{-}1)^2 + FAR^2)^{0.5}$                                      (7)

where $N_H$, $N_M$, $N_F$, and $N_N$ are the number of *hits*, *misses*, *false alarms*, and *true negatives*, respectively. A *hit* (*true negative*) refers to a condition which is classified as FLS (non-FLS; clear-sky or non-FLS-cloud) by both the algorithm and MFLs. A FLS condition (as indicated by MFLs) which is identified as non-FLS by the product is referred to as a *miss*, and the

opposite is referred to as a *false alarm*. It should be noted that one of the four cases can be true for a data point only. *POD* quantifies the skill of the method in correctly identifying the FLS cases. *FAR* indicates the portion of false alarms among all the cases classified as FLS by the algorithm. *PFD* highlights the portion of the non-FLS cases which have been correctly classified as non-FLS by the algorithm. *CSI* is a metric indicating the overall correctness of the FLS classification. *ACC* shows the portion of the cases which have been correctly identified as FLS and non-FLS. The metrics *POD*, *FAR*, *PFD*, *CSI,*

and *ACC* vary between 0 and 1. An ideal detection algorithm should have a *POD, PFD, CSI,* and *ACC* equal to 1, and *FAR* equal to 0. *BS* reflects the overall bias of the product. *BS* varies in between 0 and +∝, with the optimal value being 1. The values of *BS* greater and smaller than 1 reflect over- and under-estimation of the FLS class, respectively. As can be inferred from Eq. (6), *d* is a metric obtained by combining the statistical indicators *POD* and *FAR* and compares the performance of



the algorithm with the ideal algorithm (*FAR: 0, POD: 1*). In other words, it indicates how distant the performance of the FLS

detection algorithm is in comparison with that of the best algorithm which can ideally exist. $d$ varies between 0 and $\sqrt{2}$, with

0 being the ideal value.

The statistical indicators described above were calculated using the entire data points included in each dataset to evaluate and

compare both algorithms' overall performance over the train and four test datasets.

## 3 Validation results

Figure 2 summarizes the results obtained from validating the ML and SOFOS FLS algorithms against the MFLs

corresponding to the "train", "test₁", "test₂", "test₃" and "test₄" datasets. The statistical indicators given in this figure were

calculated using all the data points present in the five datasets (information about the exact number of data points included in

each dataset is provided in Table 1). Thus, they can be interpreted as the overall performance of the algorithms over all the

regions and time spans covered by each dataset. Validation of the ML product versus the MFLs included in the train dataset

revealed an overall classification accuracy of 0.60 and 0.98 in terms of CSI and ACC metrics for the algorithm, respectively.

The algorithm was able to correctly label about 80% of the FLS and 99% of the non-FLS (clear-sky or non-FLS-cloud)

situations included in train dataset (POD: 0.80 and PFD: 0.99). The FAR score obtained by the ML product for the same

dataset was 0.30, which resulted in a $d$ of 0.36. The FAR of 0.30 indicates that 30% of the situations labelled as FLS by the

ML product were reported as non-FLS by the ground based MFLs. Nevertheless, because of these false alarms the FLS

occurrence frequency computed for the train dataset based on the ML product is overestimated by about 14% compared to

the determination made by the MFLs (BS: 1.14). Overall, these numbers show that the ML algorithm proposed here has been

capable of detecting majority of the FLS situations over the locations and time spans included in the train dataset.

Considering that the algorithm was trained using these exact data points, it can be inferred that the algorithm has been able to

capture the subtle yet discernible distinctions between FLS and non-FLS situations (in the LIR spectral region) at locations

and time spans covered by the train dataset. This is further reinforced by comparing these error metrics with those calculated

for state-of-the-art daytime SOFOS for validation versus the same dataset. In particular, the CSI, ACC, and PFD scores

obtained by SOFOS based on validation versus the same dataset were 0.59, 0.98, and 0.98 respectively, which are close to

what was obtained by the ML algorithm. On the other hand, SOFOS showed a higher POD (0.92) compared to the ML

algorithm, which was achieved at the cost of a higher FAR (0.38), leading to larger $d$ and BS (0.38 and 1.47, respectively).

Specifically, the number of data points labelled as FLS by SOFOS was 14027 (29%) greater than what was reported as FLS

by the ML algorithm. Only about one thirds of these data points, however, were correct calls and the rest were false alarms.

In fact, all of the false alarm cases must have been identified as clear-sky by the MFLs. That is because the criteria

considered in the MFLs for screening our the non-FLS clouds is the same as the one used by SOFOS. Thus, the case of non-

FLS-cloud class included in the MFLs matches well with that of SOFOS and the validation approach presented here

essentially evaluates the skill of SOFOS in discriminating the FLS and clear-sky situation from each other. One implication





of these differences can be the fact that the ML algorithm considers a stricter criterion compared to SOFOS for classifying a situation as FLS, resulting in a more confident FLS identification. On the other hand, these misclassifications could also correspond to the cases of snow-covered land under clear sky conditions. Nonetheless, heterogeneity of topography at subpixel scale could have a strong impact on this evaluation. This is particularly important here due to the coarse resolution

of the SEVIRI pixels over Europe. Subpixel heterogeneity of topography can be problematic for validating the products as sky condition in the area covered by the SEVIRI pixel are not well represented by station measurements. Specifically, there may exist situations where the station measurements indicate a clear-sky condition, whereas majority of the area covered by a SEVIRI pixel are covered by FLS (and vice versa). Understanding the influence of topographical heterogeneity at the subpixel scale on the FAR and BS scores given here, however, can be quite challenging and is out of the scope of the current

study.

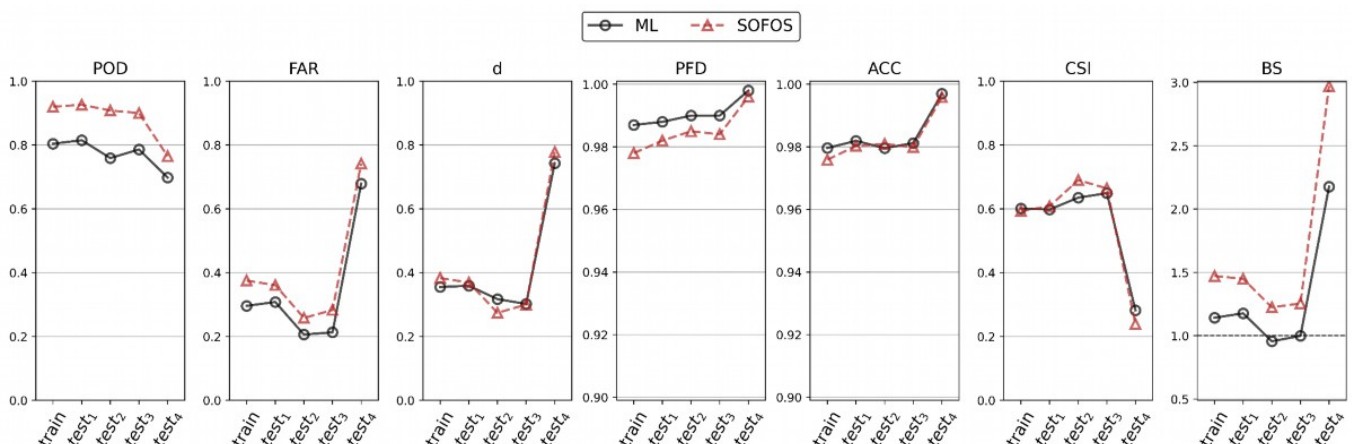

**Figure 2. Validation results of the ML (black line/circle) and SOFOS FLS detection algorithms (red line/triangle) versus the METAR FLS Labels (MFLs) for the "train", "test$_1$", "test$_2$", "test$_3$", and "test$_4$" datasets.**


Based on the preceding discussion, it can be inferred that the ML algorithm is competent and appropriate for distinguishing between FLS and non-FLS situations. This accomplishment aligns with the initial hypothesis of this study, which states that spectral LIR data can be used to differentiate FLS from cloud-free land and non-FLS clouds with comparable accuracy to existing state-of-the-art daytime FLS detection algorithms. To ascertain the applicability and generalization of the newly

proposed approach to locations and time periods not used in the training, the performance of the ML algorithm was rigorously assessed using the test$_1$, test$_2$, and test$_3$ datasets. To gauge its effectiveness, the algorithm's results were compared with those obtained for SOFOS using the same datasets (Fig. 2). These results overall suggest that the ML algorithm applicability expands beyond locations and time periods covered by the train dataset. For example, the test$_1$ dataset includes data points from the same stations as the training dataset, but it covers time periods that are not included in the training

dataset. The CSI, ACC, and PFD scores obtained by the ML model for validation versus the test$_1$ dataset were respectively



equal to 0.60, 0.98, and 0.99 which are the same as those reported for the train dataset. Nonetheless, the POD and FAR values derived from validating the ML algorithm with test$_1$ dataset were slightly higher than those obtained through validating it with the train dataset (0.82 and 0.31, respectively). Consequently, this results in a slightly increased BS (1.18), while the value of $d$ remains the same (0.36). The test$_2$ dataset evaluates the algorithm's FLS detection capability at locations

outside the train dataset but within the same time spans. On the other hand, the test$_3$ dataset evaluates the algorithm's performance at locations and time spans that it has never encountered during training. For the test$_2$ dataset, the ML algorithm achieved a CSI score of 0.64, an ACC score of 0.98, and a PFD score of 0.99. Additionally, the algorithm's POD was 0.76, with a FAR of approximately 0.21. This resulted in a $d$ of 0.32 and a BS of 0.96. Similarly, for the test$_3$ dataset, the ML algorithm achieved a CSI score of 0.65, an ACC score of 0.98, and a PFD score of 0.99. The algorithm's POD was 0.79, with

a FAR of around 0.21. This resulted in a $d$ of 0.30 and a BS of 1.00. As it can be inferred from Fig. 2, these numbers compare very well with those obtained by SOFOS for the three test datasets. In particular, ACC, CSI and PFD were merely the same for both algorithms over the three test datasets. On the other hand, the POD and FAR values obtained by SOFOS were both higher than those obtained by the ML algorithm which result in a similar $d$ but higher BS scores. Lastly, the ML algorithm was validated with the test$_4$ dataset to evaluate its performance at locations where FLS rarely occurs. This is

particularly important here because none of the two algorithms have been originally designed for such regions. Nonetheless, results obtained from this validation should be interpreted with care due to the low number of FLS cases (as identified by MFLs) included in this dataset. In particular, test$_4$ dataset only includes nearly 0.2% FLS cases and as a result, even one random misclassification can have a strong effect on the error metrics FAR, $d$, CSI and BS. For this reason, the evaluation of the algorithm in this region is done solely based on the metrics POD, PFD and ACC. As can be seen from Fig. 2, validation

of the ML (SOFOS) algorithm with the test$_4$ gave high values of POD, PFD, and ACC, being equal to 0.70 (0.76), ~1.0 (~1.0), and ~1.0 (~1.0), respectively. These numbers prove that the ML algorithm developed in the present study is applicable to regions where FLS rarely occurs.

Overall, the error metrics given above (and shown Fig. 2) show that the ML FLS detection algorithm developed here is suited for application over Europe. In particular, the algorithm features an accuracy very similar to that of the the-state-of-

the-art daytime SOFOS, but has the advantage of operating based on SEVIRI LIR bands, which allows it to be applied over the 24H of the diurnal cycle. Worth mentioning that the error metrics calculated here for SOFOS are somewhat different compared with those reported in Cermak & Bendix (2008) and Egli et al. (2018) for this algorithm, although the METAR data was used for validation purposes in the two studies. Specifically, the POD, FAR, and CSI, as reported by Cermak & Bendix (2008) for SOFOS, ranged from 0.76 to 0.83, 0.06 to 0.02, and 0.73 to 0.82, respectively. In the study by Egli et al.

(2018), the reported values of *POD* and *FAR* for SOFOS were 0.52 and 0.66, respectively. There are several reasons contributing to these differences. One reason can be the additional processing steps involved in the present study for the generation of MFLs compared with those applied in the other two studies: applying the quality flags and marking the non-FLS-cloud contaminates pixels. The difference in the location of the stations selected and the number of data points utilized for the validation highly impact the validation results. Additionally, as these error metrics are calculated as relative statistics,



the absolute number of FLS/non-FLS cases included in the datasets used for validation can make a big difference in the results obtained.

## 4 Conclusions and outlook

The main objective of the present study was to develop a diurnally stable algorithm for detection of FLS over Europe based on the observation of the SEVIRI instrument onboard MSG satellites with an accuracy comparable with that of the state-of-

the-art daytime FLS detection algorithms. The algorithm proposed here consists of a gradient boosting machine learning model (XGBoost) that is trained to classify each SEVIRI pixel as "clear-sky", "FLS", or "non-FLS-cloud" based on the SEVIRI observations in the LIR bands. Specifically, the classification is performed based on pixel values of $BT_{12.0}$, $BT_{8.7}$ - $BT_{12.0}$, $BT_{10.8}$ - $BT_{12.0}$, and $BT_{12.0}$ - $BT_{13.4}$ plus the standard deviation of each of these variables in a spatial window sized 3x3 pixels with the central pixel being the target pixel.

Six years of daytime ground-based observations from Meteorological Aviation Routine Weather Reports (METAR) at 356 European locations was utilized to generate ground truth FLS labels for training and evaluating the algorithm. Lastly, a comparison between the accuracies of the newly proposed ML and a state-of-the-art daytime FLS detection algorithm (SOFOS) was performed to address the objective of the study.

The results obtained from validating the FLS products against training and the four test datasets revealed that the ML FLS

detection algorithm proposed here is capable of discriminating the FLS from other sky conditions and its applicability in the study region (i.e., Europe) extends beyond the particular locations and time spans covered by the datapoints used for training the algorithm. Specifically, the ML algorithm features an accuracy very close to that of SOFOS over all the five datasets. The main difference between the two is in the POD, FAR, and BS achieved by them: SOFOS features a slightly higher POD compared to the ML algorithm. On the other hand, it suffers from a higher FAR and BS compared to the ML algorithm. One

implication of these differences can be the fact that the ML algorithm considers a more restrict criteria for classifying a situation as FLS, resulting a more confident FLS class. Another advantage that the ML algorithm has over SOFOS is that it operates solely based on the SEVIRI channels in the LIR spectral region, which allows it to be applicable over the 24H of the diurnal cycle. In addition, the ML FLS detection algorithm presented here is efficient in terms of computation time. This makes the algorithm very suitable for operational purposes such as monitoring, nowcasting, and forecasting of FLS as well

as reprocessing the historical SEVIRI data. Furthermore, as the algorithm is the first of its kind that is a single algorithm applicable over day and night over Europe, it can potentially provide new insights into the FLS life cycle over Europe and help enhancing the performance of existing FLS forecast products. This will be of particular use in the development of short-term forecasts of FLS dissipation for applications such as photovoltaic power forecasting.

Nonetheless, it should be noted that there are limitations associated with the validation procedure performed here: Although

the algorithm presented here classifies the states of the sky into the three classes clear-sky, FLS and non-FLS-cloud, the validation procedure presented here only evaluates its skill in differentiating the FLS situations from the clear-sky and non-



FLS-cloud. Its skill in discriminating between clear-sky and non-FLS-cloud situations yet remains to be unvalidated due to limited information obtained from METAR observation. Further, although the ML algorithm presented here operates based on SEVIRI LIR bands, we have not been able to provide a quantitative evaluation of its accuracy over nighttime as the procedure applied here for the identification of the non-FLS-clouds falling withing SEVIRI's FOV consisted of thresholding the SEVIRI channels falling within the limits of the solar spectrum. Additionally, all the datapoints used in the present study for algorithm evaluations are located over land and limited to European region. For this reason, the algorithm's applicability over water and at other regions is not quantitatively validated. Nevertheless, diurnal stability and applicability of the algorithm over water as well as its ability to discriminate non-FLS-clouds from clear-sky situation were confirmed through visual inspection of several sequences of satellite images and the corresponding resulting classification maps. See the supplement animation (S1) for more information. In this animation, the left-hand panel shows a false-color RGB image constructed based on the SEVIRI raw channel data with the red, green, and blue channels being $BT_{12.0}$ - $BT_{13.4}$, $BT_{8.7}$ - $BT_{12.0}$, and $BT_{10.8}$ - $BT_{12.0}$, respectively. In this panel, the green color represents the high clouds, and the light and dark red colors represent the clear-sky and FLS, respectively. The right-hand panel of this animation also shows the outputs of the ML FLS detection algorithm developed in the present study.

In addition to the limitations discussed above, there are also some degrees of uncertainty involved with the validation performed here, which originate from the spatio-temporal matching of the SEVIRI pixels with the METAR locations, heterogeneity of the land surface topography and sky condition at the subpixel level, and the difference in the measurement devices and calibration standards used in generating the METAR observations. Nevertheless, the encouraging results obtained in the present study show that the spatio-temporal matching of both datasets has been very good, and the pre-processing methods applied have successfully screened out many of the ambiguous situations.

In summary, the FLS detection algorithm developed here is accurate and as it can be applied over the 24H cycle of the day it can provide new insights to FLS studies and enable development of existing/new FLS fore-/now-casting techniques. The results obtained here, and the potential of the algorithm encourages an expansion of the validation to other regions covered by SEVIRI instrument. Ideally, the algorithm presented here could also be further developed to discriminate between fog and low stratus, detect FLS layers that are placed underneath optically thin non-FLS-clouds, or/and differentiate snow-covered and snow-free lands from each other.




## Appendix A

Figure A1 schematically describes the process followed in Sect. 2 for creation of the MFL (Modified FLS Labels) included the datasets "train", "test₁", "test₂", "test₃", and "test₄".

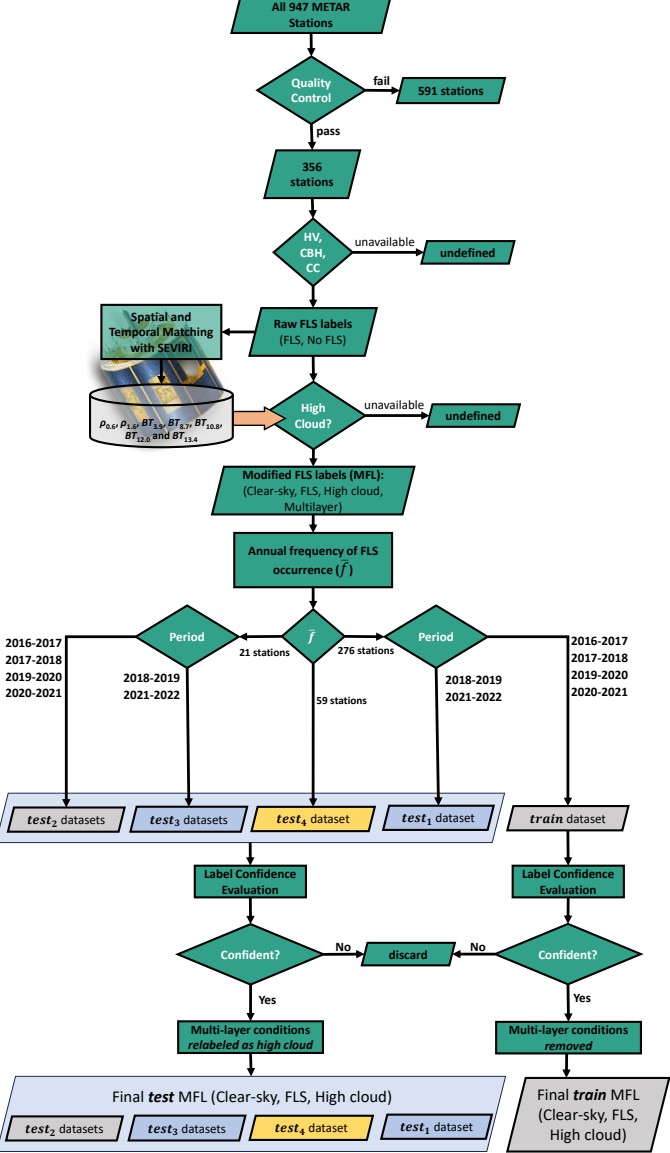

**Figure A1. Schematic description of the process followed in Sect. 2 for creation of the MFL (Modified FLS Labels) included the datasets "train", "test₁", "test₂", "test₃", and "test₄".**



**Code availability**

The python implementations of SOFOS and the FLS detection model developed here can be provided upon request. The source codes for the XGBoost python implementations used in the present study can be freely obtained from https://xgboost.readthedocs.io/en/stable/index.html, last access: 13.10.2023. The satpy python package can be freely accessed from https://satpy.readthedocs.io/en/stable/, last access: 13.10.2023.

**Data availability**

All the raw data used in the present study are publicly available, and details on the datasets are provided in Sect. 2. The Supplement animation S1 can be downloaded from https://doi.org/10.5281/zenodo.10244714.

**Author contribution**

All authors contributed to the review of the manuscript. JC, JF, and SK contributed to the conceptualization of the project. JF, SK, JC and TZ contributed to the design of the study and interpretation of the results. JF contributed to the project administration. MZ downloaded and organized the raw SEVIRI data and applied the SOFOS algorithm to them. BJ and SK generated of the label data. BJ contributed to the design of the study, performed the computations, interpreted the results and wrote the manuscript.

**Competing interests**

The contact author has declared that none of the authors has any competing interests.

**Acknowledgement**

This study was funded by the German Federal Ministry of Economic Affairs and Climate Action (BMWK; project ID: 03EE1083A and 03EE1083C).

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
