# Peer review of "Algorithm for continual monitoring of fog based on geostationary satellite imagery"

_EGUsphere, 2023_

## Author Comment (AC1)

**Manuscript number:** EGUSPHERE-2023-2885
**Old title:** Algorithm for continual monitoring of fog life cycles based on geostationary
satellite imagery as a basis for solar energy forecasting
**New title:** Algorithm for continual monitoring of fog life cycles based on geostationary
satellite imagery
**Submitted to:** Atmospheric Measurement Techniques (AMT)

Dear anonymous reviewer,

On behalf of all authors, we would like to kindly thank you for your very constructive
comments and suggestions, and as well for spending your precious time on reviewing our
manuscript.

Please find below our answers to your inquiries.

Note that the list of references mentioned in this document is given on the last page.

Sincerely yours,
Babak Jahani, Jan Cermak

**Referee #1**

**Summary**

- The authors try to develop a IR-only fog detection using channels at 8.7, 9.7, 10.8, 12, 13.4 microns of SEVIRI. The best combinations of these channels for fog detection are inferred by XGBoost. Before this work, the same group developed another fog detection algorithm, named SOFO, based on multiple tests (cloud, ice, snow, droplets) using IR-vis channels at 0.6, 0.8, 1.6, 3.9, 8.7, 10.8, 12 microns from the same satellite instrument. SOFO is more physically based but requires more channels to operate. They show that the XGBoost-based method has a POD of ~75-80% and an FAR of ~20-30%.
- The authors argue that their XGBoost-based method is expandable because while their validation/test datasets contain regimes that are not included in the training dataset, the POD and FAR performances on those regimes remain comparable to the training (except test4 which has fog frequencies < 1%).
- Overall the presentation is clear. The motivation of developing a new method of IR-only fog detection is discussed. The steps creating the training dataset, test1, test2, test3, and test4 are outlined. These details make sure their work is reproducible.

**Comments**

**C#1.1**

My biggest concern is that the FAR is well above 20%. Statistically, a method would be deemed useful if it has a FAR less than 5%-10%. My concern also applies to their previous method, SOFOS, which has an even greater FAR (as high as 40%). The XGBoost-based method may seem to be better but note that the POD of XGBoost-based is 10% less than SOFOS. So based on the POD and FAR, in my opinion, both methods do not seem to be practical. A potential problem is that a part of the training dataset has been based on the SOFOS method to create the fog/low stratus labels. Therefore, errors in SOFOS would propagate into the training dataset and eventually upset the training of XGBoost.

**A#1.1**

Thank you for raising these valuable points. Please let us clarify this matter:
1. You are correct that the XGBoost-based algorithm shows a lower FAR, but also a lower POD compared to SOFOS, which translates into a similar accuracy level for both products. In fact, this is in line with the hypothesis and objective of the study: "spectral LIR data can be used to differentiate FLS from cloud-free land and non-FLS clouds with comparable accuracy to an existing state-of-the-art daytime FLS detection algorithm". This fact has been highlighted in the revised manuscript at lines 64-66, 81-84 (Section 1), 351-353 (Section 4) and at lines 432-434 (Section 5). Here we are not aiming at proposing a technique that has better accuracy compared to SOFOS. Instead, the novelty here is to have a technique that is applicable over day and night, as well as the day-night transition times, and has an accuracy comparable to an existing technique making use of a wider part of the

electromagnetic spectrum (i.e. visible-range channels). The idea here is not to develop a perfect technique for all situations, but one that works around the clock.

2. As the error metrics *POD*, *FAR*, *CIS* and *BS* are calculated as relative statistics, the absolute number of FLS/non-FLS cases included in the datasets used for validation can make a big difference in the results obtained. This is particularly important here because they are calculated relative to a small subset of the data: the FLS-positive cases (as identified by the truth or predicted product) which are inherently low in number. As a result of the relatively small denominator, they can show a relatively high degree of sensitivity to a few misclassifications. Additionally, they do not provide a global image about the overall classification accuracy of the product, as they do not account for the "*true negative*" instances, which are very large in number for FLS. For these reasons, although the metrics *POD*, *FAR*, *CIS* and *BS* provide essential and detailed information on the product's performance, they need to be interpreted with care. To account for these two matters the error metrics *ACC* and *PFD* were introduced. As the denominator of these metrics are rather large and take the "*true negative*" instances into consideration, they are expected to be better suited for showing the product's overall performance over the whole dataset. As can be seen from Figure 1 presented in the manuscript, both products show comparably good performances in terms of these two metrics over all datasets (ACC and PFD are both above 0.97). This argument has now been included in the revised manuscript at lines 385-395 (Section 4).

3. Another matter to point out is that the METAR data used in algorithm evaluation itself does not perfectly capture ground truth. One important aspect is that METAR data are taken from the ground, looking up, whereas the satellite takes the opposite perspective. Plus, the METAR stations belong to different organizations, managed by different organizations and do not use unified instrumentations. Additionally, as mentioned in the manuscript, what is observed from a point ground measurement does not necessarily match with what can be seen by a space-born observer that has a pixel size of 16 to 64km$^2$ over Europe. Please see sections 3.1, 3.2, and lines 475-480 of the revised manuscript where the resulting problems are discussed. We have tried minimizing the effect of these variables by imposing the quality control protocols described in the sections 2.3 and 3.2 of the revised manuscript and correcting for the non-FLS clouds passaging through the satellite's line of sight (described in Section 3.1 of the revised manuscript) -this is especially relevant in Europe, where about 30% of FLS events are obscured by other clouds (Cermak, 2018). Of course, this cannot be perfect and has its own limitations, leading to the leak of unwanted datapoints to the training and test sets.

We hope this response addresses your concerns. Thank you for prompting us to clarify these important points.

Please be advised that in line with this comment and C#2.1 (first comment from referee #2), we have added a new figure to the manuscript (i.e., Figure 3) which shows 1) applicability and consistency of the algorithm over day and night as well as during day-night transitions, 2) applicability over water and land/water transition regions, and 3) consistency of the classifications with the observed radiances.

Furthermore, for the same objectives we had provided an animation presented in the supplementary material S1 (downloadable from https://zenodo.org/records/10244714) in the initial submission. In this animation, the left-hand panel shows a false-color RGB image constructed based on the SEVIRI raw channel data with the red, green, and blue channels being $BT_{12.0}$ - $BT_{13.4}$, $BT_{8.7}$ - $BT_{12.0}$, and $BT_{10.8}$ - $BT_{12.0}$, respectively. In this panel, the green color represents the high clouds, and the light and dark red colors represent the clear-sky and FLS, respectively. The right-hand panel of this animation also shows the outputs of the ML FLS detection algorithm developed in the present study.
* * *
**C#1.2**

There is a lack of physical explanation why BT12.0, BT8.7 - BT12.0, BT10.8 - BT12.0, and BT12.0 - BT13.4 would have been "chosen" by XGBoost.  Their searching process (randomizing the combinations of the channels and find which minimal set of combinations give a desirable result) is typical of modern machine learning approaches. But in applied sciences, the interpretation of the results is as important as the method itself.

**A#1.2**

Thank you for pointing it out. Very brief information as provided in the initial submission (at lines 271-274 in the initial submission). In line with your comment, the explanations were further extended. Additional explanations were added to Section 3.4 of the revised manuscript at lines 269-281.
* * *
**C#1.3**

Most of the discussions of PV in the abstract and the text are irrelavant to the study. At least the discussions of PV in the abstract should be removed.

**A#1.3**

In line with your comment, the discussions of PV in the abstract were removed.
* * *
**C#1.4**

In addition, the term "life-cycle" in the title is misleading because the current manuscript does not study the life-cycle of fog/low stratus.

**A#1.4**

In line with your comment, the title has been modified to "Algorithm for continual monitoring of fog based on geostationary satellite imagery".
* * *
**C#1.5**

A "train dataset" should be a "training dataset".

**A#1.5**

The suggested change has been applied to the manuscript.
* * *
**References**

Cermak, J.: Fog and Low Cloud Frequency and Properties from Active-Sensor Satellite Data, Remote Sens., 10, 1209, https://doi.org/10.3390/rs10081209, 2018.

---

## Author Comment (AC2)

**Manuscript number:** EGUSPHERE-2023-2885
**Old title:** Algorithm for continual monitoring of fog life cycles based on geostationary satellite imagery as a basis for solar energy forecasting
**New title:** Algorithm for continual monitoring of fog life cycles based on geostationary satellite imagery
**Submitted to:** Atmospheric Measurement Techniques (AMT)

Dear anonymous reviewer,

On behalf of all authors, we would like to kindly thank you for your very constructive comments and suggestions, and as well for spending your precious time on reviewing our manuscript.

Please find below our answers to your inquiries.

Note that the list of references mentioned in this document is given on the last page.

Sincerely yours,
Babak Jahani, Jan Cermak

**Referee #2:**

**Summary**

- This paper presents a method of low stratus and fog (FLS) detection day and night time detection. The novelty is to train a machine learning algorithm (XGBOOST) using open observations from airport weather stations (METAR)  and Meteosat Second Generation IR observations.
- The general methodology is robust and the results are sufficiently interpreted.

**Comments**

**C#2.1**

The position of paper considers as state-of-the-art only "the well-established and well-validated SOFOS". Which is a daytime technique. However, the SAF NWC products, operationnal since 2016, provides FLS detection day and night. The Optimal Cloud Analysis proposes also a day night cloud top height product. We can also cite MSG-CPP from KNMI and APOLLO-NG recently developped by the DLR. Even if SOFOS is certainly an excellent reference to test this new algorithm, this paper cannot ignore the state-of-start.  For a complete analysis, if a day and night product exists, authors must compare their algorithms with them, not only from SOFOS which has been developed within the same team.

**A#2.1**

Thank you for the insightful comments and suggestions. We appreciate your emphasis on comparing our technique with relevant methods, and we agree that our study can benefit from comparison with external FLS products. In response, we actively sought out a SEVIRI-based FLS classification product; however, we found that none of the mentioned products directly fit our study requirements.
Here are the main reasons:

- **NWC-SAF:** as mentioned on the webpage https://www.nwcsaf.org/, mainly provides software packages plus some output product images (as a reference to what users can obtain), rather than public products. Generating comparable data would require implementing their algorithm with user-specific settings and auxiliary data from ECMWF, leading to non-standardized results.
  For this reason, although algorithm for deriving "cloud type" is presented in the software package manual (https://www.nwcsaf.org/Downloads/GEO/2021.2/Documents/Scientific_Docs/NWC-CDOP3-GEO-MF-CMS-SCI-UM-Cloud_v2.0.3.pdf),  and the example outputs are visualized in https://www.nwcsaf.org/web/guest/nec/geo-geostationary-archive, based on our best knowledge, this data is not publicly available for order to on any EUMETSAT related platforms.
  Additionally, it should be noted that although this software can be applied over day and night, it does not mean that it is the same algorithm applied over night and day: as mentioned in Section 3.3 of

https://www.nwcsaf.org/Downloads/GEO/2021.2/Documents/Scientific_Docs/NWC-CDOP3-GEO-MF-CMS-SCI-UM-Cloud_v2.0.3.pdf, this software uses the SEVIRI channels "R0.6µm R1.38µm T3.8µm T7.3µm T8.7µm T10.4µm T10.8µm T12.0µm" for cloud classification.  Among them, "R0.6µm R1.38µm T3.8µm" are solar zenith angle  dependent, which are taken out or affected by the intensity of daylight. The changes taken place in the classification due to algorithm switch can be clearly observed in the visualized outputs of the software in https://www.nwcsaf.org/web/guest/nec/geo-geostationary-archive (a good example can be seen when inspecting the following dates: 21-11-2023, 23-11-2023, 27-11-2023, …).

- **MSG-CPP from KNMI** (https://msgcpp.knmi.nl/): This product provides cloud mask, top height, temperature, phase, and other cloud properties, but it does not include cloud classification or FLS-specific information. These products are retrieved based on the NWC-SAF software and ECMWF model outputs. Cloud top height could be a relevant dataset at first glance, but it by itself cannot be used for a comparison. In fact, deriving FLS data from this product would necessitate developing a new method based on cloud-top height thresholds to be defined, which would again involve us as developers, limiting the benefit of an external comparison.

- **EUMETSAT Optimal Cloud Analysis** product (https://navigator.eumetsat.int/product/EO:EUM:DAT:MSG:OCA): similar to MSG-CPP provides cloud related data, but no cloud classification/type information is present in this product. Thus, it does not include FLS classification, limiting its applicability to our work.

- **DLR APOLLO-NG:** Unfortunately, we could not locate publicly available data for this product.

- **EUMETSAT Cloud Top Height (CTH):** Although this product includes cloud height information, it cannot be directly interpreted as a FLS mask. The manual of this product even clearly states that extracting fog information would require future enhancements in Section 10.3.2.3. Therefore, deriving FLS data from this product would necessitate developing a new method by defining and applying threshold values, which would again involve us as developers, limiting the benefit of an external comparison. Additionally, it should be noted that a first step to cloud top height determination in this approach is cloud detection, which similar to the products mentioned above, comprises of separate day- and night-time approaches.

Given these points, SOFOS remains the most suitable reference for our purposes, as it provides standardized outputs and aligns with the objectives of our study. We hope this addresses the rationale behind our comparison choices.

In line with this comment and #C1.1 (first comment from referee #1), we have added a new figure to the manuscript (i.e., Figure 3) which shows 1) applicability and consistency of the algorithm over day and night as well as during day-night transitions, 2) applicability over water and land/water transition regions, and 3) consistency of the classifications with the observed radiances.

Furthermore, for the same objectives we had provided an animation presented in the supplementary material S1 (downloadable from https://zenodo.org/records/10244714) in the initial submission. In this animation, the left-hand panel shows a false-color RGB image constructed based on the SEVIRI raw channel data with the red, green, and blue channels being $BT_{12.0}$ - $BT_{13.4}$, $BT_{8.7}$ - $BT_{12.0}$, and $BT_{10.8}$ - $BT_{12.0}$, respectively. In this panel, the green color represents the high clouds, and the light and dark red colors represent the clear-sky and FLS, respectively. The right-hand panel of this animation also shows the outputs of the ML FLS detection algorithm developed in the present study.

- - - - - - - - - - - - - - - - - - - - - - - - - - - - - - - - - - - - - - - - - - - - - - - - - - - - - - - -

**C#2.2**

Reading this paper was quite difficult. An effort should be made to shorten and simplify argumentations.

**C#2.2.1**

Especially, paragraph 2.4 is extremely long and should be shortened with the help of the very clear Table 1. Moreover 2.4 should divided in several sub sections (e.g FLS label characteristics can be separated from dataset building.

**A#2.2.1**

Thank you for your suggestion. Section 2 was divided into two sections (i.e., section 2: data, and section 3: methods). The section named as 2.4 in the original submission was split into 3 sections: "2.4 FLS labels", "2.5 FLS flag quality assessment", and "2.6 Training and testing datasets". The section named as "Machine learning algorithm" in the original submission was also split into two sections, namely "Machine learning algorithm" and "Feature selection".

Please note that we had also included Figure A1 in the initial submission to facilitate the reading and understanding of the FLS label generation process.

- - - - - - - - - - - - - - - - - - - - - - - - - - - - - - - - - - - - - - - - - - - - - - - - - - - - - - - -

**C#2.2.2**

Paragraph 2.5 shows some useless repetition such as the list of channel and channel combination which is written twice (and a third time in the conclusion paragraph).

**A#2.2.2**

Section 2.5 has now been revised. We would rather be keeping information in the conclusion for the readers who would like to read the abstract and jump to the conclusions.

To make the section naming more consistent with the content, conclusions sections was renamed to "Summary, Conclusions and outlook".

- - - - - - - - - - - - - - - - - - - - - - - - - - - - - - - - - - - - - - - - - - - - - - - - - - - - - - - - - - -

**C#2.2.3**

Lines from 304 to 320 could be synthetised in a table (this is just a suggestion)

**A#2.2.3**

Equations and their explanations were moved to appendix (B).
The naming of the confusion matrix related terms was modified, and their descriptions were synthesized in a table to facilitate the reading and understanding.

- - - - - - - - - - - - - - - - - - - - - - - - - - - - - - - - - - - - - - - - - - - - - - - - - - - - - - - - - - -

**C#2.2.4**

Lines 374-380 repeat arguments already  in paragraph 2.4

**A#2.2.4**

In line with your suggestion, the mentioned lines were removed.

- - - - - - - - - - - - - - - - - - - - - - - - - - - - - - - - - - - - - - - - - - - - - - - - - - - - - - - - - - -

**C#2.2.5**

Lines from 385 to 390 are a simple repetition of information visible in figure 2.

**A#2.2.5**

In line with your suggestion, the mentioned lines were removed.

- - - - - - - - - - - - - - - - - - - - - - - - - - - - - - - - - - - - - - - - - - - - - - - - - - - - - - - - - - -

**C#2.3 Detailed remarks :**

**C#2.3.1**

Line 85 and 109 : centered instead of "cantered"

**A#2.3.1**

Corrected!

- - - - - - - - - - - - - - - - - - - - - - - - - - - - - - - - - - - - - - - - - - - - - - - - - - - - - - - - - - -

**C#2.3.2**

Line 114 please give the reference (website) to find METAR data

**A#2.3.2**

Reference to the METAR data is
https://mesonet.agron.iastate.edu/request/download.phtml
We have updated the "data availability" section accordingly.

- - - - - - - - - - - - - - - - - - - - - - - - - - - - - - - - - - - - - - - - - - - - - - - - - - - - - - - - - - -

**C#2.3.3**

Line 145 : precise in which EUMETSAT product did you find elevation above sea level ?

**A#2.3.3**

Thank you for noticing. It was obtained from EUMETSAT LSA-SAF auxiliary data at
https://lsa-saf.eumetsat.int/en/user-support/auxiliary-data/, last access:

16.01.2023. We have updated the figure caption and the "data availability" section accordingly.

- - - - - - - - - - - - - - - - - - - - - - - - - - - - - - - - - - - - - - - - - - - - - - - - - - - - -

**C#2.3.4**

Line 194 : why only per winter year ?

**A#2.3.4**

The summer months (i.e., June, July, and August) were excluded from the analysis because the FLS occurrence frequency is very low over these months (Egli et al., 2017) and their inclusion in the analysis largely affects the imbalance between the observed FLS positive and negative cases. This argument has been outlined in the revised manuscript. Please see lines 110-113 of the revised manuscript.

- - - - - - - - - - - - - - - - - - - - - - - - - - - - - - - - - - - - - - - - - - - - - - - - - - - - -

**C#2.3.5**

Line 214 : why did you choose to do a quality check after division the datasets ? There is risk of unbalanced dataset size.

**A#2.3.5**

This was done because the goal was to sample data based on different FLS regimes. Whereas quality checks were applied mainly to flag data points that are likely to represent homogeneous sky conditions. This explanation was added to the revised manuscript in Section 3.3 at lines 245-248.

- - - - - - - - - - - - - - - - - - - - - - - - - - - - - - - - - - - - - - - - - - - - - - - - - - - - -

**C#2.3.6**

Line 244 : test3 (0.2%) is certainly test4

**A#2.3.6**

Thanks for noticing. Corrected!

- - - - - - - - - - - - - - - - - - - - - - - - - - - - - - - - - - - - - - - - - - - - - - - - - - - - -

**C#2.3.7**

Line 260 What is the "previous model" ?

**A#2.3.7**

Here, we are referring to the regression tree ensembles. The term "previous model" was changed to "previous regression tree" to clarify the matter.

- - - - - - - - - - - - - - - - - - - - - - - - - - - - - - - - - - - - - - - - - - - - - - - - - - - - -

**C#2.3.8**

 Line 444 ":Although" with lower case after ":"

**A#2.3.8**

Corrected!

- - - - - - - - - - - - - - - - - - - - - - - - - - - - - - - - - - - - - - - - - - - - - - - - - - - - -

**References**

Egli, S., Thies, B., Drönner, J., Cermak, J., and Bendix, J.: A 10 year fog and low stratus climatology for Europe based on Meteosat Second Generation data, Q. J. R. Meteorol. Soc., 143, 530–541, https://doi.org/10.1002/qj.2941, 2017.

---

## Author Response (AR2)

**Manuscript number:** EGUSPHERE-2023-2885
**Title:** Algorithm for continual monitoring of fog based on geostationary satellite imagery
**Submitted to:** Atmospheric Measurement Techniques (AMT)

Dear editor and anonymous reviewers,

Once again, on behalf of all authors, we would like to kindly thank you for your very constructive comments and suggestions, and also for spending your precious time on reviewing our manuscript.

Please find below our answers to your inquiries.

Note that
   a) list of references mentioned in this document is given on the last page.
   b) due to an update in the reference manager used, all in-text and the bibliography fields included in the manuscript had to be updated. These changes are marked in the manuscript with track changes.

Sincerely yours,
Babak Jahani, Jan Cermak

**Referee #1**

**Comments and suggestions**

**C#1.1**

I appreciate the very detailed responses by the authors.

The revised manuscript is much better written and organized.

I agree that the POD and the FAR are sensitive to the fraction of fog/low stratus cases in the dataset, which is only 3% in this study. So a 30% of FAR means 1% of misidentified fog/low stratus cases in the whole dataset. Is 1% a big number? In terms of the absolute number, 1% relative to the whole dataset is small. But in terms of the predictive power, for every 4 reported fog cases, 1 of them is a false alarm, which is not small relative to reported fog/low stratus cases. In addition, a 80-90% POD means that 0.3-0.6% of the whole data which are fog/low stratus cases cannot be detected. In the absolute sense, 0.3-0.6% (relative to the whole dataset) is small, but10-20% (relative to the fog/low stratus cases) is not small. In this sense, the authors' response by introducing ACC and PFD does not address my comment about the high FAR directly.

In any case, it was not my purpose to debate how the verification statistics should be interpreted. Actually, with the current results, I think it is fine for the authors report their detection algorithm, but they should say a few words on how the accuracy of their detection method can be further improved (e.g. how can the FAR be reduced?) in the future.

The authors have added some discussions (Lines 190-200 and Lines 455-477) about the limitations of ground station data, such as the potential errors due to the spatiotemporal matching with the satellite pixels and the big difference between the spatial resolutions of 3x3 km2 satellite pixel versus a point measurement at the ground, which may have contributed to a part of the FAR. But again, I don't think these discussions would directly address the issue of the high FAR. Such limitations of ground station data are not unique to fog/low stratus detection; the same limitations are applicable to any comparison between satellite measurements and ground measurements (e.g. of greenhouse gases, surface temperature, etc). It is more important that the authors discuss about the potential deficiencies of the current detection algorithm besides the potential problems of the validation procedure.

**A#1.1**

Thank you for the clarification and the opportunity to account for this. In line with your suggestion, we have included some discussion on this topic in Sections 4 and 5. In summary, we have analyzed False Positive (FP; see Appendix B) and False Negative (FN; see Appendix B) predictions of the ML algorithm across the five datasets to gain insights on the deficits of the algorithm, and proposed ideas on how these deficit can be accounted for in the future.

In particular we have added the following in Section 4 (Results and discussion) at lines 395-415 of the revised manuscript:

"

Figure 3 provides detailed insights into the false-negative and false-positive (see Appendix B) predictions of the FLS ML technique across the five datasets. From Figure 3(a) it can be seen that the majority (~80-90%) of pixels falsely classified as FLS by the ML algorithm (i.e., false alarms) were actually clear-sky. Similarly, Figure 3(b) shows that ~83-92% of false-negative pixels (i.e., undetected FLS pixels) were predicted as clear-sky by the ML algorithm. These results suggest that the algorithm has room for improvement in distinguishing between clear-sky and FLS. Indeed, finding it challenging for the algorithm to separate clear-sky from FLS (especially in the complex geo- and atmospheric conditions of Europe) is expected, because FLS emits at temperatures close to the Earth's surface, resulting in only small differences between the LIR spectral signatures of clear-sky and FLS conditions. Although the algorithm shows some limitations in distinguishing between clear-sky and FLS, its performance is acceptable for meeting the objectives of the study. Especially, given that it is a lightweight approach relying solely on calibrated MSG-SEVIRI LIR data. Nevertheless, its accuracy can be further enhanced by incorporating auxiliary datasets. For example, integrating digital elevation maps (e.g., as in Egli et al., 2018), and surface or atmospheric temperature and humidity information from external sources (e.g. from reanalysis as in NWC SAF, 2019) could improve *POD* and *FAR*. Additionally, applying pixel- or spatial-based pre-/post-processing steps, such as those used in Pauli et al., (2022) and Andersen and Cermak, (2018) may help better identify FLS events and correct potential misclassifications, leading to a decreased *FAR*. These enhancements, however, fall outside the scope of the present study and are left for future work.

[Figure]

Figure 3. Detailed information on false-positive and false-negative (see Appendix B) predictions of the FLS ML technique across training, test1, test2, test3, test4 datasets. Panel (a) shows the percentage of false-positive instances that were clear-sky and non-FLS-cloud contaminated according to the ground truth dataset (i.e., MFL). Panel (b) shows the percentage of false-negative instances that were categorized as clear-sky and non-FLS-cloud by the ML algorithm.
"

And the following in Section 5 (Summary, Conclusions and outlook) at lines 480-485:

"

While the algorithm achieves the accuracy required for this study, a closer examination of its false-positive and false-negative (see Appendix B) predictions across the five datasets suggests that it could be improved in distinguishing between clear-sky and FLS. Its accuracy can be enhanced by incorporating auxiliary datasets (e.g. digital elevation maps and surface/atmospheric temperature and humidity information from external sources) and pixel- or spatial-based pre-/post-processing steps.

"

- - - - - - - - - - - - - - - - - - - - - - - - - - - - - - - - - - - - - - - - - - - - - - - - - - -

**Referee #2:**

**Comments and suggestions**

**C#2.1**

In A#2.1 You agree that your study can benefit from comparison with external FLS products. However, your demonstrate that other products such as SAFNWC output, msg cpp algorithm or APOLLO-NG are not comparable to your results. Thank you for this long and documented answers.

However, you still did not mention any of these products and continue to show SOFOS as "the well-established and well-validated".

I have no doubt on the quality of SOFOS, but such article cannot ignore the state of the art. SOFOS is not the only attempt to derive cloud low cloud data from Meteosat Second Generation. Moreover, the mention "well-establised" is not so appropriate. In which community is it "well established" ? Is there any operational weather service using SOFOS ?

I ask to the authors to cite more works on similar or related low cloud property retrieval from MSG satellite to inform the reader that author works and its associated reference SOFOS are valuable works among others.

Other comments have been well answered, modifications on the text have been successfully done.

**A#2.1**

We describe SOFOS as an "established" algorithm in that it has been used as the basis for numerous scientific studies (about 10-20) since its first publication. It has been implemented by Deutscher Wetterdienst (DWD) staff into the satpy framework as fogpy as a basis for nowcasting applications. However, we do not insist on the label "well-established" and have replaced it by "existing" and discarded the term "state-of-the-art" where appropriate in the revised manuscript. Please see the Manuscript with track changes at lines  22, 88, 125, 315, 337, and 474.
Inline with your suggestion, the following references are now mentioned in the revised manuscript (please see lines 45, 60, and 410):
- NWC/GEO (NWC SAF, 2019),
- APOLLO-NG (Klüser et al., 2015),
- (Fuchs et al., 2022),
- MSG-CPP (we were not able to locate a reference for it) , and
- (Pauli et al., 2024)

- - - - - - - - - - - - - - - - - - - - - - - - - - - - - - - - - - - - - - - - - - - - - - - - - - - - - -

**References**

Andersen, H. and Cermak, J.: First fully diurnal fog and low cloud satellite detection reveals life cycle in the Namib, Atmos. Meas. Tech., 11, 5461–5470, https://doi.org/10.5194/amt-11-5461-2018, 2018.

Egli, S., Thies, S., and Bendix, J.: A hybrid approach for fog retrieval based on a combination of satellite and ground truth data, Remote Sens., 10, https://doi.org/10.3390/rs10040628, 2018.

Fuchs, J., Andersen, H., Cermak, J., Pauli, E., and Roebeling, R.: High-resolution satellite-based cloud detection for the analysis of land surface effects on boundary layer clouds, Atmos. Meas. Tech., 15, 4257–4270, https://doi.org/https://doi.org/10.5194/amt-15-4257-2022, 2022.

Klüser, L., Killius, N., and Gesell, G.: APOLLO-NG – A probabilistic interpretation of the APOLLO legacy for AVHRR heritage channels, Atmos. Meas. Tech., 8, 4155–4170, https://doi.org/10.5194/amt-8-4155-2015, 2015.

NWC SAF: Algorithm Theoretical Basis Document for Cloud Product Processors of the NWC/GEO (v2.1), https://doi.org/https://www.nwcsaf.org/Downloads/GEO/2018/Documents/Scientific_Docs/NWC-CDOP2-GEO-MFL-SCI-ATBD-Cloud_v2.1.pdf, 2019.

Pauli, E., Cermak, J., and Andersen, H.: A satellite-based climatology of fog and low stratus formation and dissipation times in central Europe, Q. J. R. Meteorol. Soc., 148, 1439–1454, https://doi.org/10.1002/qj.4272, 2022.

Pauli, E., Cermak, J., Bendix, J., and Stier, P.: Synoptic Scale Controls and Aerosol Effects on Fog and Low Stratus Life Cycle Processes in the Po Valley, Italy, Geophys. Res. Lett., 51, https://doi.org/10.1029/2024GL111490, 2024.